# An Unpowered Knee Exoskeleton for Walking Assistance and Energy Capture

**DOI:** 10.3390/mi14101812

**Published:** 2023-09-22

**Authors:** Xinyao Tang, Xupeng Wang, Yanmin Xue, Pingping Wei

**Affiliations:** 1School of Mechanical and Precision Instrument Engineering, Xi’an University of Technology, Xi’an 710048, China; 2Research Center for Civil-Military Integration and Protection Equipment Design Innovation, Xi’an University of Technology, Xi’an 710054, China; 3State Key Laboratory of Mechanical Manufacturing System Engineering, Xi’an Jiaotong University, Xi’an 710043, China

**Keywords:** unpowered exoskeleton, knee joint, walking assistance, energy capture, energy compensation mechanism

## Abstract

In order to reduce the energy consumption of human daily movement without providing additional power, we considered the biomechanical behavior of the knee during external impedance interactions. Based on the theory of human sports biomechanics, combined with the requirements of human–machine coupling motion consistency and coordination, an unpowered exoskeleton-assisted device for the knee joint is proposed in this paper. The effectiveness of this assisted device was verified using gait experiments and distributed plantar pressure tests with three modes: “not wearing exoskeleton” (No exo.), “wearing exoskeleton with assistance “ (Exo. On), and “wearing exoskeleton without assistance” (Exo. Off). The experimental results indicate that (1) This device can effectively enhance the function of the knee, increasing the range of knee movement by 3.72% (*p* < 0.001). (2) In the early stages of the lower limb swing, this device reduces the activity of muscles in relation to the knee flexion, such as the rectus femoris, vastus lateralis, and soleus muscles. (3) For the first time, it was found that the movement length of the plantar pressure center was reduced by 6.57% (*p* = 0.027). This basic principle can be applied to assist the in-depth development of wearable devices.

## 1. Introduction

Human energy conversion is an extremely complex process. Walking is a simple process with relatively low energy consumption, but, in daily life, more energy is used for walking than any other activity [1]. Different wearable exoskeletons have been developed for human enhancement. Their purpose is to improve the performance of the human body, reducing energy consumption, delaying fatigue, and increasing body speed. Wearable exoskeleton devices have a high development space in medical rehabilitation, power assistance and military fields [2,3]. One example is the IHMC exoskeleton or BLEEX [4,5]. This lower limb exoskeleton is designed based on ergonomics by simulating human motion. It can improve human walking durability and load-bearing capacities through mechanisms, controls, and sensors, enabling wearers to work with greater weight under the same conditions [6], further reducing human fatigue and increasing exercise time.

It has been proved that a walking-assisted exoskeleton (WAE) can effectively reduce the workload required in walking. However, the limitations of a portable power supply and large and bulky actuators limit the use of these devices. For these reasons, underactuated and passive (unpowered) exoskeletons are an attractive option. Passive WAE has been developed to achieve energy-saving walking. By using only passive components and driven by manpower, the power demand is reduced to zero. For passive exoskeletons, their lightweight design is more important. To reduce the energy consumption of the wearer, many different strategies have been adopted. Some designs have focused on weight support systems [7], while others use complex spring systems to save energy by reducing muscle demand [8,9]. Tang et al. [10] summarized how assist components used by unpowered exoskeleton-assist robots usually implement a tension spring [11,12,13,14], a torsion spring [8,15], pneumatic artificial muscle [16,17,18], a constant force spring [19], a scroll spring [20,21], a single (double) spring [22,23,24], a spring lever and elastic cable. The existing evaluation methods of exoskeletons mainly include metabolic Chen Ben, muscle activation rate [25], joint torque, joint angle, plantar pressure, and joint force. Liu et al. [26] studied the influence of weight distribution within knee exoskeleton components on simulated muscle activity during three functional movements. Ranaweera et al. [27] developed a passive knee exoskeleton robot, which could store the energy generated by the buckling and recycle it when helping with standing-up. The research and development of unpowered exoskeleton power-assisted robots could solve the energy and power problems that limit the development of traditional power-assisted robots and reduce the control difficulty and manufacturing costs in the robotic research and development process [28].

It can be concluded from the literature above that the passive assisted exoskeleton of the lower limb in the human body can not only be used as the external skeleton, but also as wearable equipment, with good wearability to provide protection or assistance for the wearer. Further research and optimization both at home and abroad could effectively solve the problem of a secondary energy supplement, which this device requires, while also improving its energy conversion efficiency. The purpose of this study is to analyse the biomechanical characteristics and energy change characteristics of lower human limbs. An unpowered exoskeleton assist device for the knee joint is proposed, and gait experimental data collection and analysis are carried out. The performance of the exoskeleton is evaluated through experimental tasks to assist people in walking. It can be applied to healthy people and hemiplegic patients to assist with leg lifting so as to reduce energy consumption during leg exercises and walking. The second part of this paper showcases the design principle of this exoskeleton model. The third part introduces the mathematical model of the human exoskeleton system. The fourth part describes and discusses the experimental design, data processing and results analysis. In the fifth part, we draw the conclusion of this paper.

## 2. Biomechanical Model of Human–Machine Coupling Movement

### 2.1. Kinematic Characteristics of Lower Limbs

Human walking requires the coordination of multiple muscles and joints, moving the legs at the same time in a periodic manner, which aims to support weight and maintain dynamic stability. When external conditions change, muscles produce new strength to maintain walking. Using data from the lower limb joint driving torque and joint angle [29] as the input, the biomechanical power was obtained by multiplying joint moment by the joint angular velocity. The average knee strength was negative, which means that the knee or lower limb muscles consume metabolic energy to absorb kinetic energy. Therefore, the ideal method is to use the negative work period of the knee joint to reduce the metabolic cost of human movement.

Normal human gait usually follows a general rule: legs are repeatedly lifted and land in the air. When the right (left) heel begins to touch the ground and the heel touches the ground again, this period is called a gait cycle, which is usually expressed in seconds. Figure 1 it the continuous stages after normalizing the time of a gait cycle. The initial contact time (IC) occurs in the process of the lower limb from swinging to standing, starting when the right (left) heel touches the ground. In the midstance (MS), bodyweight is borne by the right (left) lower limbs and moves from the heel to the forefoot. At the terminal stance (TS), the load-bearing surface of the right (left) plantar decreases gradually and prepares to lift the right (left) heel to step on the ground and leave the ground. During the pre-swing (PS), the right (left) side load-bearing foot is only in contact with the ground, and the left (right) side foot is at the moment when the heel touches the bottom. In the initial swing (IS), the early stage is when the right (left) side support foot leaves the ground. In mid-swing (MS), the right (left) lower limb retracts and approaches the left (right) lower limb, and the distance between the feet reaches the minimum value of the gait cycle. At the terminal swing (TS), the right (left) lower limbs step forward and returns to the initial gait posture.

By analyzing the structure of lower limbs, their working joints can be divided into the hip, knee, and ankle. Each moving joint can be used as the object of energy collection. The gravity balance state is shown in Figure 2. It focuses on the movement of the lower limb and simplifies the upper limb. The red rod represents the upper limb, the blue rod represents the right lower limb, the green rod represents the left lower limb, the hollow circle represents the movable joint, and the solid circle represents the center of mass for the rod. During walking, the human body is constantly undertaking positive and negative work to overcome gravity. The direction of gravity G at the shank is vertically downward, and the direction of speed v is perpendicular to the shank and points to the direction of motion. The angle α is the angle between the direction of gravity G and the direction of velocity v. The work performed by lower limb joints can be described using three cases: (1) at α<90°, the lower limbs are stretched and perform positive work (Figure 2a); (2) at α=90°, the lower limbs stand upright without performing work (Figure 2b); (3) at α>90°, the lower limbs perform flexion and negative work (Figure 2c).

### 2.2. Human-Machine Coupling Energy Model

From the perspective of energy capture in lower human limbs. When human lower limb joints perform negative work, the energy consumed is reduced through energy compensation. Therefore, when part of the energy during the positive work of the moving joint is compensated for negative work, it can reduce the energy consumption of the lower limbs during negative work and achieve the purpose of auxiliary walking. To sum up, the energy capture device stores energy when the joint performs negative work. The energy capture device releases energy when the joint undertakes positive work.

In Figure 3, the fitting curve of average knee power in a gait cycle is shown. Six typical postures of the knee joint are selected at the top of Figure 3 (see Table 1) and compared with the average power fitting curve at the bottom. Within this figure, the horizontal *X*-axis represents a complete gait cycle (%), the vertical *Y*-axis represents the knee power (W), the blue line represents the original average power curve $P_{a}$, and the red line represents the expected average power curve $P_{e}$ after assistance. The difference between these two fitting curves is represented by the shadow composed of 45° oblique lines, and the shadow area above and below the horizontal axis is represented by $S_{1}$ and $S_{2}$ respectively.

Then the region S composed of $S_{1}$ and $S_{2}$ can be expressed as
(1)$$S=\int {|P_{e}-P}_{a}|dt$$

In the process of continuous walking, the energy capture device constantly switches back and forth between the energy storage state and the energy release state. When the moving joint is performing positive work, the energy capture device needs to store energy $P_{s}$, part of which comes from the energy of $P_{s1},$ which is generated by absorbing positive work; the other part, $P_{n},$ is additionally provided by the human body:(2)$$P_{s}=P_{s1}+P_{n}$$
(3)$$\left\{\begin{array}{c} P_{s1}<P_{s\prime },\;P_{n}=0,\;s_{1}<0\\ P_{s1}=P_{s\prime },\;P_{n}=0,\;s_{1}=0\\ P_{s1}>P_{s\prime },\;P_{n}>0,\;s_{1}>0 \end{array}\right.$$
(4)$$P_{s}=\sum_{0}^{+w}{|P_{e}-P}_{a}|$$
where $P_{n}=0$ indicates that the human body does not need to provide additional energy, and $P_{n}>0$ indicates that the human body needs to provide additional energy.

When the moving joint is performing negative work, the energy $P_{s2}\;$($P_{s1}\geq P_{s2}$) released by the energy capture device is used to compensate the negative work energy $P_{r}$ of the moving joint, and additional energy $P_{m}$ is required to undertake the negative work after the moving joint obtains energy compensation:(5)$$P_{r}=P_{s2}+P_{m}$$
(6)$$\left\{\begin{array}{c} P_{s2}<P_{r},\;P_{m}>0,\;s_{2}>0\\ P_{s2}=P_{r\prime },\;P_{m}=0,\;s_{2}=0\\ P_{s2}>P_{r\prime },\;P_{m}=0,\;s_{2}<0 \end{array}\right.$$
(7)$$P_{r}=\sum_{0}^{-w}{|P_{e}-P}_{a}|$$
where $P_{m}=0$ means that the moving joint does not need to consume additional energy to perform negative work and $P_{m}>0$ means that the moving joint needs to consume additional energy to undertake negative work. As shown in Figure 4, there may be energy changes in the process of assisting people to walk.

It can be observed that the magnitude of energy absorbed and released by the energy capture unit directly determines whether the purpose of assisted walking can be achieved, as shown in Table 2.

From the perspective of using the device, in a gait cycle, knee joint unpowered assist walking devices should be synchronized with the periodic movement of lower human limbs. In the process of lower limb walking, the changes in limb motion parameters include the joint angle, force, moment and plantar pressure. This power-assisted device can combine the movement of the mechanical structure with the biomechanical characteristics of human sports so as to convert mechanical energy into electrical energy or other forms of energy and feed back to the human body. In the process of lower limb walking, the changes in limb motion parameters include the joint angle, force, moment, and plantar pressure et al. This assisted walking device can combine the movement of the mechanical structure with the biomechanical characteristics of human sports so as to convert mechanical energy into electrical energy or other forms of energy and feedback to the human body. For example, the ankle exoskeleton designed and developed by Carnegie Mellon University mentioned above, through the change in the ankle angle during walking, means that the energy storage spring deforms to produce elastic potential energy, realizing the process of converting mechanical energy into elastic potential energy.

It is worth mentioning that energy capture device needs to meet two basic working states, including storing and releasing energy. In the design process of energy capture devices, a locking mechanism is added to minimize the energy consumed by the moving joint when undertaking negative work so as to ensure minimum stress on the lower limbs when standing vertically and obtain a high-performance energy capture device.

## 3. Energy Model of Energy Capture Device

### 3.1. Conservation of Energy of Human Walking

In the new design of the exoskeleton, scroll springs are used for gravity compensation. According to the energy conservation theorem, in order to realize gravity compensation during walking, the total potential energy of the human exoskeleton system, including spring and gravity, must be constant. Figure 5a shows the coupling model of the human exoskeleton system.

Figure 5b shows the human exoskeleton system model during walking. In this model, the leg is simplified as a double link mechanism representing the thigh and shank, the foot is approximately the point at the end of the lower leg, represented by $m_{3}$. In order to fully compensate for gravity, the total potential energy needs to be constant in all configurations. The expression of total potential energy is given by the following:(8)$$E=E_{S1}+E_{S2}+E_{G}$$
where $E_{S1},{\;E}_{S2}$ and $E_{G}$ represent the gravitational potential energy of spring 1, spring 2 and the human leg, respectively.

The zero potential energy surface *S* is defined by the hip joint $O_{1}$, and the gravitational potential energy of the human exoskeleton system can be obtained by the following formula:(9)$$E_{G}=E_{thigh}+E_{shank}+E_{feet}$$
where $E_{thigh}$, $E_{shank}$, and $E_{feet}$ represent the gravitational potential energy of the thighs, legs and feet, respectively.
(10)$$E_{thigh}=-m_{t}gl_{t}^{\prime }\cos\theta_{1}-m_{1}gl_{1}\cos\theta_{1}$$
(11)$$E_{shank}=-m_{s}g[l_{s}^{\prime }\cos\left(\theta_{2}-\theta_{1}\right)+l_{t}\cos\theta_{1}]-m_{2}g[l_{2}\cos\left(\theta_{2}-\theta_{1}\right)+l_{t}\cos\theta_{1}]$$
(12)$$E_{feet}=-m_{f}g[l_{s}\cos\left(\theta_{2}-\theta_{1}\right)+l_{t}\cos\theta_{1}]$$
where $m_{i}(i=1,2,3)$ represents the mass of the i-th link of the human leg, $m_{aj}(j=1,2)$ represents the mass of the j-th link of the exoskeleton leg, $l_{j}$ represents the length of the j-th link, $l_{j}$ and $s_{j}$ represent the distance from the mass center (COM) of the j-th link for the human leg and the exoskeleton leg to the j-th joint, respectively. $\theta_{1}$ and $\theta_{2}$ represent the rotation angle of the hip and knee respectively.

From Formula (9) to Formula (12), this can be obtained simultaneously.
(13)$$E_{G}=A\cos\theta_{1}+B\cos\left(\theta_{2}-\theta_{1}\right)$$

Among them, the values of parameters A and B are as follow. $A=-m_{t}gl_{t}^{\prime }-m_{1}gl_{1}-(m_{s}+m_{2}+m_{f})gl_{t}$, $B=-m_{s}gl_{s}^{\prime }-m_{2}gl_{2}-m_{f}gl_{s}$.

### 3.2. Elastic Potential Energy of Scroll Spring

The scroll spring model is shown in Figure 5c. When the O-point of the scroll spring in the free state coincides, the working parts of the spring box and the non-spring box are completely the same, but the number of idling turns when reaching the working part is different. When the spring is in the working part, the deformation angle φ and torque T of the scroll spring show linear variation characteristics [30]. Therefore, the deformation angle φ of the scroll spring is directly proportional to the bearing torque T, i.e.,
(14)$$\varphi =\frac{T}{K}$$
where K is the stiffness of the scroll spring.

The tail end of the scroll spring is fixed, and this fixed point is point A, which is r from the center O of the scroll spring. When the torque T is applied to the shaft, point A is affected by the torque $T_{1}$, tangent force $P_{t}$ and radial force $P_{r}$, as shown in Figure 5c. If the infinitesimal ds spring element body is taken along the spring length s, the spring deformation energy dU in this element is
(15)$$dU=\frac{T^{2}d_{s}}{2EI}$$
where E is the elastic modulus of the material, I is the moment of inertia for the material section, and T is the torque acting on this section. When the effective length of the scroll spring is l, the deformation energy U of the spring can be obtained by integrating it along the full length of the curve. The calculation is as follows:(16)$$U=\int_{0}^{l}dU=\int_{0}^{l}\frac{T^{2}}{2EI}d_{s}=\frac{T^{2}l}{2EI}$$

After the torque T is applied on the shaft, the scroll spring bears the same torque in all sections for the whole length, which is equal to the torque applied on the shaft. According to the card theorem in mechanics: the rate of change in the strain energy U for the elastic rod to a certain position P on the rod is equal to the position δ corresponding to the load P, which is expressed by the formula δ=∂U/∂P. Therefore, the deformation angle φ of the scroll spring can be obtained as follows:(17)$$\varphi =\frac{\partial U}{\partial T}=\frac{Tl}{EI}$$

The expression of torsional stiffness K for the scroll spring obtained from Formulas (14) and (17) is as follows:(18)$$K=\frac{EI}{l}$$

In addition, another expression for the scroll spring energy storage formula obtained from Formulas (16) and (17) is as follows:(19)$$U=\frac{1}{2}T\varphi$$

If the same two scroll springs are selected in this paper, the elastic potential energy is $E_{s1}=E_{s2}=U$ for spring 1 and spring 2.

The deformation angle is represented by the number of deformation turns n. Because φ=2πn, the number of working turns in the scroll spring is as follows:(20)$$n=\frac{Tl}{2\pi EI}$$

### 3.3. Parameter Design and Modeling of Scroll Spring

In the following numerical calculation, the participant is an adult male with a height of 1.75 m and a weight of 65 kg. The simulation process for applying the spring mechanism only to the human body is carried out. When a participant walked at a selected speed wearing only an exoskeleton with a spring mechanism, the knee moment changed with the stiffness of the scroll spring. Xie et al. [21] found that wearing the exoskeleton further changed the biomechanical power of the knee joint. When the total stiffness of spring was about 5 N·m·rad^−1^~6 N·m·rad^−1^, the performance of the exoskeleton was best. In this paper, the energy storage structure of the double coil spring was adopted, and the appropriate rigidity of scroll spring was 2.5 N.m rad^−1^~3 N.m rad^−1^. Finally, the stiffness of scroll spring was K = 3 N.m rad^−1^.

The size selection and performance calculation of the coil spring were carried out according to the JB/T7366-1994 [31] design and calculation of the plane scroll spring. This standard is only applicable to coil springs of rectangular section materials with a thickness of 0.5–4 mm and width of 5–80 mm. In addition, in order to meet the wearing requirements of the human body, the knee joint should be worn reasonably in the sagittal plane, and the outer diameter of the coil spring should be less than or equal to 50 mm. In this study, a 60Si2Mn spring steel strip with a thickness h of 0.8 mm and width b of 10 mm was selected as the material of the plane scroll spring in the device. Its elastic modulus E was 206 Gpa, tensile strength $\sigma_{b}$ was 1863 Mpa, density ρ was 7850 kg/$m^{3}$, and the length l of the spring was 1000 m.

The coil spring is installed in the coil spring box, the tail end is fixed, and the fixed point A is in contact with the surface of the coil spring box. The O-point at the center of the coil spring is fixed with the rotating shaft. It is inserted into the rotating shaft in the way of clearance fit, and the drive of the lower leg deforms. Then, the limited torque $T_{j}$ that the scroll spring can bear is as follows:(21)$$T_{j}=\frac{bh^{2}}{6}\sigma_{b}$$

The maximum output torque $T_{2}$ of the scroll spring is as follows:(22)$$T_{2}=K_{3}T_{j}=\frac{1}{6}K_{3}bh^{2}\sigma_{b}$$

Among them, $K_{3}$ is the coefficient selected by different fixing methods. When hinged fixing, $K_{3}\in [0.65,0.70]$, when pin fixing, $K_{3}\in [0.72,0.78]$, when V-type fixing, $K_{3}\in [0.80,0.85]$, and when lining fixing, $K_{3}\in [0.90,0.95]$. This is calculated as $T_{2}\approx 1.689\;N\cdot m$.

The diameter $d_{1}$ of the mandrel wound by the scroll spring can be described as
(23)$$d_{1}\geq \left(15-25\right)h$$

In this study, $d_{1}=0.012\;m$ was selected. The scroll spring is tightly wound on the mandrel under load, and the outer diameter $d_{2}$ of the spring coil can be determined as
(24)$$d_{2}=\sqrt{\frac{4lh}{\pi }+d_{1}^{2}}$$

It can be calculated that $d_{2}\approx 0.034\;m$. The inner diameter of the spring box where the scroll spring is placed is expressed by $D_{2}$, and the inner diameter of the coil when the scroll spring is loose is expressed by $D_{1}$, which is obtained by the following:(25)$$D_{2}=\sqrt{2.55lh+d_{1}^{2}}$$
(26)$$D_{1}=\sqrt{D_{1}^{2}-\frac{4lh}{\pi }}$$

$D_{2}\approx 0.047\;m$ and $D_{1}\approx 0.034\;m$ are calculated by Formulas (25) and (26). The scroll spring is placed in the spring box, and the number of turns $n_{1}$ without the torque is
(27)$$n_{1}=\frac{1}{2h}\left(D_{2}-\sqrt{D_{2}^{2}-\frac{4lh}{\pi }}\right)=\frac{1}{2h}\left(D_{2}-D_{1}\right)$$

The number of turns $n_{2}$ of the scroll spring on the mandrel is calculated as follows:(28)$$n_{2}=\frac{1}{2h}\left(\sqrt{\frac{4lh}{\pi }+d_{1}^{2}}-d_{1}\right)=\frac{1}{2h}\left(d_{2}-d_{1}\right)$$

Through the calculation of Formulas (27) and (28), $n_{1}=7.875$ cycles and $n_{2}=13.813$ cycles were obtained.

According to the basic parameters of the scroll spring and its maximum output torque $T_{2}$, the theoretical working revolution $n_{t}$ of the scroll spring can be calculated as
(29)$$n_{t}=\frac{6T_{2}l}{\pi Ebh^{3}}=\frac{K_{3}l\sigma_{b}}{\pi Eh}$$

Therefore, the theoretical working revolution of the spring n≈3.0601. The effective working revolution $n_{e}$ of the scroll spring is calculated as follows.
(30)$$n_{e}=K_{4}(n_{2}-n_{1})$$
where $K_{4}$ is the effective coefficient, and the corresponding value can be selected through the correlation according to the ratio of $d_{1}$ and n. Then $n_{e}=5.225$ turns. The stored energy U of the scroll spring can be obtained according to Formula (19).
(31)$$U=\frac{1}{2}T\varphi =\frac{1}{2}T_{2}\times 2\pi n=T_{2}\pi n_{t}$$

The energy storage density of the scroll spring refers to the energy storage per unit mass, which can be expressed by the ratio of energy U to mass m combined with formula (17), and the energy storage density q of scroll spring can be obtained.
(32)$$q=\frac{U}{m}=\frac{1}{\rho V}T_{2}\pi n_{t}$$

Therefore, U≈16.230 J and q≈258.441 J/kg are calculated using Formulas (31) and (32).

When the deformation angle φ is 60°, the bearing torque $T_{p}\approx 3.142\;N\cdot m$, the effective energy storage $U_{p}\approx 1.645\;J$ and the effective energy storage density $q_{p}\approx 26.193\;J/kg$ are calculated according to Formulas (21), (31) and (32). After calculation, the design parameters of the scroll spring are shown in Table 3.

### 3.4. The Exoskeleton Model

Based on the above theory, an unpowered knee exoskeleton for assisted walking and energy capture was designed, and two prototypes were made. Their practical feasibility was tested with resin and the effectiveness of the design was tested using an aluminum alloy. According to the wearer’s preliminary feedback on the resin prototype (Figure 6a), the aluminum alloy prototype (Figure 6b) was manufactured with a mass of 1.15 kg.

Figure 6 shows an overall view of the knee exoskeleton worn on the human body. The fixation of the exoskeleton is mainly composed of two parts. Coil springs are installed at the thigh and knee joints, respectively, to assist with inelastic deformation. These two parts are attached to the rigid support frame and connected with the flexible part to prevent wear on the human body. Coil spring 1 is driven by the movement of the shank, and coil spring 2 is driven by an inelastic cable called the transmission cable connected with the movable shaft of coil spring 1.

The knee joint installation part includes a lower leg flexible support, a rigid support, a transmission shaft with an embedded coil spring (at coil spring 1), and a rotating ring with an inelastic cable (at coil spring 1). The drive shaft, rotating ring, and coil spring are coaxial with the knee joint. For the thigh installation part, including the ratchet and dial, a transmission shaft with an embedded coil spring (at coil spring 2), and a rotating ring with an inelastic cable (at coil spring 2) are all arranged in the coaxial direction. In addition, a pawl is arranged, which can better fit the normal gait of the human body with the ratchet wheel and give play to the power-assisted role of another coil spring. The drive cable connects the two rotating rings so that the motion of the knee joint can be further transmitted to the thigh. When the exoskeleton works, rotating the jacking screw and fastening screw in the ring can prevent the cable from sliding so as not to affect the power-assisted process of the coil spring.

During walking, the range of motion in the lower limbs is very large. If the exoskeleton is installed away from the center of gravity in the human body, it may bring additional metabolic costs to the wearer. Based on this consideration, it is proposed that it be installed near the mass center of the user’s thigh and shank to minimize this potential additional metabolic cost. As shown in Figure 3, the knee joint exoskeleton includes a thigh mounting part and a knee mounting part, which are connected through a rigid structure. Figure 6c is a schematic diagram of a human body wearing the exoskeleton. Figure 6d,e, provides the details of the thigh and knee joint, respectively.

When the knee joint is extended, the displacement of the thigh and shank drives the transmission shaft (at coil spring 1) and a rotating ring with an inelastic cable (at coil spring 1). Then, the transmission cable rotates the rotating ring at the coil spring to drive the other coil spring (at coil spring 1) and the ratchet pawl.

When knee flexion occurs in the swing phase, the elastic energy stored in the knee mounting part (at coil spring 1) is released, and the mechanical restoring force helps the lower limb muscles lift the shank. The elastic energy in the spring of the thigh mounting part (at coil spring 2) is also released. During a complete gait cycle, coil spring 1 helps the muscles slow down and accelerate the knee joint, while coil spring 2 only provides similar help during deceleration.

As is well known, in walking activities, the knee joint angle variable is the largest, mainly occurring in the sagittal plane. In this study, spiral springs were selected as energy storage components, which could repeatedly provide a greater recovery force through elastic deformation. After one practice, it returned to its original position. One end is fixed by a coil spring box clamp, and the other end is installed by a coaxial insert. As the angle of the knee joint changes, the coil undergoes elastic bending deformation, causing the coil spring to twist in its own plane. This torque is used to provide power. During exercise, skeletal muscles provide a source of strength. Tendons connect muscles and bones, and muscles contract to produce joint movement. In addition, springs simulate human muscles, while rigid structures simulate the tendons to provide connections, enabling the spring to generate potential deformation energy, ultimately and effectively reducing the wearer’s own energy consumption. During this process, certain limitations inevitably arise due to the preloading force set by the coil spring, as shown in Figure 7. The advantage is that these limitations do not affect the body’s movement posture.

Through these mechanisms, when the muscle consumes metabolic energy for negative work, the knee joint exoskeleton absorbs energy. When muscles consume metabolic energy for active work, the knee joint exoskeleton provides potential elastic energy. Therefore, the knee exoskeleton can partially replace the function of the lower limb muscles and reduce the metabolic cost of human walking.

## 4. Performance Evaluation

### 4.1. Experiment Preparation and Process

The proposed knee exoskeleton provides an elastic element parallel to the knee during the stance phase, but the quality and attachment mode of exoskeleton attachment affect the body state at all times. The added mass of the exoskeleton has a gravitational effect on the hip’s extensor and knee flexor muscles because they must lift the exoskeleton mass during the early swing. In addition, this increased mass also has an inertial effect on the gluteal flexors because they must accelerate the mass during the swing. Finally, without some restrictions, it is difficult to achieve attachment to the body, which can limit the range of motion and cause discomfort. To assess whether exoskeletons represent an effective solution to reduce metabolic costs during weight-bearing walking, we evaluated the performance of knee exoskeletons under three conditions, including “not wearing exoskeleton”(No exo.), “wearing exoskeleton with assistance “(Exo. on), and “wearing exoskeleton without assistance”(Exo. off). The third case (Exo. off) was evaluated to assess the limitations associated with the additional quality represented by the load-bearing device itself, which is an important consideration in the design of such systems. In all trials, the exoskeleton was worn on the right leg. Figure 7 shows the experimental flow, data analysis and process in a flow chart.

In this experiment, the participant was a healthy adult male (age 23 years, height 175 cm, weight 62 kg), without joint, skeletal muscle, nerve or other diseases, and no history of surgery. According to the experimental guidelines, we informed the participant of the purpose and detailed process of this experiment. After obtaining the consent of the participant, the gait acquisition experiment was carried out.

A three-dimensional (3D) motion acquisition and analysis system (NOKOV Motion Capture System, Beijing, China), distributed plantar pressure test system (JASENCO, JSP-C5, Beijing, China), and wireless surface electromyography test equipment (Delsys) were used in the experiment. The motion capture system included motion capture cameras (10), three-dimensional force measuring platforms (3), computers, marking points, correction frames and other auxiliary equipment, as shown in Figure 8a. The shooting frequency of the motion capture camera was 100 Hz, and the data collection frequency of the force plate was 1500 Hz. The participant carried reflective marker balls and captured and each marker point was recorded on the human body by reflecting light of the same wavelength to the camera. Then, through the motion analysis software, the required motion data were collected. The distributed plantar pressure test system included acquisition equipment, data line, computer, and data acquisition software, as shown in Figure 8b.

The improved engineering prototype was used in this experiment, as shown in Figure 6b. During the experiment, the kinematics and dynamics of the joints were measured. The participant’s lower limbs used a modified Cleveland Clinic pelvis and leg marker set (left and right ASIS, left and right greater trochanter, left and right PSI, public marker points, thigh marker points, medial and lateral epicondyle, lower leg marker points, medial and lateral malleolus, calcaneus, foot, fifth metatarsal) (Figure 8c). After the static standing test was calibrated, Visual 3D (CMotion Inc, Germantown, MD, USA) modeling software was used to reconstruct the joint kinematics, dynamics, and centroid trajectory, assuming that the left and right legs were symmetrical.

Before the test, the participant was informed of the specific experimental procedure and was trained, using the device for at least 30 min each day. The participant first trained on open ground, and then put on a fall-resistant seat belt on a treadmill. In this training course, for safety reasons, the selected participant’s gait had to be wide enough to prevent collision between supports, and the knee extension ensured that the locking mechanism worked when standing.

In the three-dimensional motion acquisition experiment (Figure 9a), in order to avoid the marker ball from being blocked by clothes, the participant wore tight shorts and was shirtless to improve the accuracy of experimental data. Before the start of the walking experiment, it was required that static data be collected from the participant. The participant stepped on a three-dimensional force measuring board with their legs separated, and kept their arms extended. Then, the participant was asked to walk in a straight line along a prescribed route of about 5 m according to their own habits, and step on the three force plates with their two legs one after another. During each walk, the participant adjusted their pace according to the metronome rhythm, in order to ensure that the experimental pace was uniform. The participant walked straight from the force plate and returned to the starting point of the previous walk, which counted as one walk cycle. Gait segmentation was achieved by measuring the ground reaction forces. Each walking task collected 50 walking cycles, with a rest interval of about 2 min between each walking cycle. Large fluctuations were avoided in collected kinematic and dynamic data due to experimental fatigue. We tried to ensure that the participant’s natural gait was retained during the experiment.

During the plantar pressure dynamic test (Figure 9b), the participant was required to walk through the plantar pressure measuring board using their left foot, and then walk back using their right foot so the dynamic plantar pressure distribution of both feet could be obtained. The specific experimental process was as follows. First, the participant walked straight ahead at their chosen speed for 10 m, with the equipment in the middle. Then, while walking on the pressure board, the participant ensured that the index finger of their foot was on the marked white line. Finally, three successful experiments were collected for each dynamic task with an interval of 15 min between the two experiments to eliminate the effect of fatigue.

In the process of a human wearing experiment, the exoskeleton can be evaluated more comprehensively by analyzing muscle activity. We considered the function of the muscle, the depth of the muscle surface, and the position of the muscle. Delsys wireless surface electromyography test equipment was used in the experiment. The rectus femoris (RF), vastus lateralis (VL), semitendinosus (SEM), tibialis anterior (TA), peroneus longus (PL), and soleus (SOL) muscles were selected for the test and analysis to evaluate the assistance effect of the exoskeleton (Figure 9c). First, hair on the skin’s surface was removed using a depilator. Secondly, the skin was wiped with 75% medical alcohol. Finally, the skin’s surface was polished to remove the cuticle. The direction of electrode placement was parallel to the direction of muscle fiber, and the electrode was fixed using electrode paste. In order to prevent the EMG block from loosening and falling off during the exercise, the EMG block was wrapped with a skin membrane and the surface EMG signal was recorded.

### 4.2. Data Processing of Human Gait

Based on the background of anatomical concepts, the three basic planes and three basic axes of human motion (see Figure 10) are briefly explained. The three basic planes are the sagittal plane, frontal plane, and horizontal plane [32]. The three basic axes are sagittal, frontal, and vertical. Among them, the frontal plane is composed of the coronal axis and vertical axis, which divides the human body into its front and back parts. The transverse plane is composed of the sagittal axis and coronal axis, which divides the body into its upper and lower parts. The sagittal plane is composed of the sagittal axis and vertical axis, which divides the body into its left and right parts, and it is also the only symmetrical plane. In the normal walking process of the human body, walking movement mainly occurs in the sagittal plane, and there is little to no movement in the frontal and horizontal planes. Therefore, when designing a lower limb-assisted exoskeleton robot, the flexion/extension motion of each joint is the primary goal for consideration. On this basis, we only selected effective data of the sagittal motion for processing and analysis.

Some motion capture recordings proved unusable due to technical difficulties associated with the loss or migration of motion capture markers, as well as spurious markers due to reflections from the exoskeleton. Therefore, the exact timing of the steps used varied among participants, and it was not possible to analyze 50 steps for all the trials. After processing 50 groups of gait data, 15 groups of normal and available gait files were selected for data analysis. In general, at least one minute after the start of the test, the earliest available reconstruction was used to minimize the impact of fatigue.

As shown in Figure 6, the experimental flow and data analysis process are shown on the right. The experimental data were collected using the NOKOV dynamic capture test system, and the gait data (.c3d format) file was the output. Combined with Mokka software (version 0.6.2), the preliminary data analysis of the marked points was performed to screen out the complete gait test data of the marked points. Importing the gait data file into Visual 3D could extract plantar pressure information and select the gait motion to be analyzed during walking. The gait experimental analysis in this paper selected the gait test data from the first landing of the right foot to the second landing of the right heel so as to obtain changes in the knee angle, force and moment. It is worth mentioning that plantar pressure changes on the pressure plate then extracted plantar pressure data on the output table, and finally processed the plantar pressure change curve in the gait cycle.

### 4.3. Statistical Methods

All statistical tests were performed using the statistical program SPSS (Version 26), and the significance level of all analyses was set at *p* < 0.05. The one-way repeated measure analysis of variance (ANOVA) of three modes (No exo., Exo. on, and Exo. off) was used to verify the effect of equipment on gait and plantar pressure. And two modes (No exo. and Exo. on) of the one-way repeated measurement analysis of variance (ANOVA) were used to verify the average muscle activation of the device for the whole stride. The least significant difference (LSD) post hoc test was performed to determine the difference between the conditions with the statistically significant main effects determined using ANOVA.

## 5. Result and Discussion

### 5.1. Gait Experiments

The participant walked on the ground at a speed of their choice and simultaneously tested their walking test under three conditions, including No exo., Exo. on, and Exo. off. The gait data from the first landing of the right heel to the second landing of the right heel were intercepted. The intercepted gait data of lower limbs were normalized to 101 data points.

Figure 11 shows the changes in the hip knee power of participants in walking tasks, including changes in their hip knee power from the first heel’s contact to the second heel contact of healthy participants in three walking tasks (Figure 11a); a box diagram of the change range in hip knee power for a complete gait cycle is also shown (Figure 11b). It can be clearly seen from Figure 11a that the first peak, second valley and third valley of knee power in the “Exo. on” state were less than those in the “Exo. off” state and were reduced by about 41.55%, 25.83% and 6.87% respectively. During the second peak and the first valley of knee power, the state of “Exo. on” increased by about 16.31% and 94.65%, respectively, compared to the state of “Exo. off”, and the third peak was almost the same. It can be clearly seen from Figure 11b that for the state of “Exo. on”, the variation range in the hip and knee power was greater than that of “Exo. off”, and the average value of the variation range in hip and knee power increased by about 21.13% (*n* = 39 groups; one-way repeated-measures ANOVA; F = 8.355, *p* = 0.001; LSD post hoc analysis: *p* = 0.005) and 39.07% (*n* = 39 groups; one-way repeated-measures ANOVA; F = 18.609, *p* < 0.001; LSD post hoc analysis: *p* < 0.001), respectively. For the “Exo. off” state, the variation range in hip and knee power was less than that of “No exo.”, and the average value of the variation range in hip and knee power was reduced by about 21.60% (LSD post hoc analysis: *p* < 0.001) and 21.69% (LSD post hoc analysis: *p* < 0.001), respectively.

Figure 12 shows the joint angle of the participant in the walking task. Figure 12a to Figure 12c show the curves for the hip angle and knee angle from the first heel’s contact to the second toe’s contact for healthy participants in the three walking tasks. For the task of “Exo. on”, the motion range of the knee joints was greater than that of the “Exo. off”, which increased by about 3.72% (*n* = 39 groups; one-way repeated-measures ANOVA; F = 14.379, *p* < 0.001; LSD post hoc analysis: *p* < 0.001). For the task of “Exo. off”, the range of motion in the knee joints was smaller than that of the task of “No exo.”, which was reduced by about 2.05% (LSD post hoc analysis: *p* = 0.005), respectively.

Figure 13 shows the changes in the hip and knee moment of the participant during the walking tasks, including the changes in the hip and knee moment from the first heel’s contact to the second heel’s contact for the healthy participant in three walking tasks (Figure 13a), and a box diagram for the variation range in hip and knee moment in a complete gait cycle (Figure 13b). It can be clearly seen from Figure 13a that the first peak value of the knee moment in the “Exo. on” state was less than that in the “Exo. off” state, which was reduced by about 8.58%; however, the second peak value of the knee joint moment, the “Exo. on”, state increased by about 25.24% compared with the “Exo off” state. As can be clearly seen from Figure 13b, for the “Exo. on” state, the variation range in the hip moment was greater than “Exo. off”, and the average joint moment increased by about 9.82% (*n* = 39 groups; one-way repeated-measures ANOVA; F = 6.983, *p* = 0.003; LSD post hoc analysis: *p* = 0.027); however, when the variation range for the knee moment was less than “Exo. off”, the average joint moment decreased by about 2.12% (*n* = 39 groups; one-way repeated-measures ANOVA; F = 0.166, *p* = 0.848; LSD post hoc analysis: *p* = 0.662). For the “Exo. off” state, the joint moment variation range for the hip and knee was less than that of “No exo.”, and the average joint moment of the hip and knee was reduced by about 13.61% (LSD post hoc analysis: *p* = 0.001) and 0.48% (LSD post hoc analysis: *p* = 0.920), respectively. This insignificant result indicates that wearing exoskeletons does not affect the knee moment.

Overall, in the dynamic gait experiment, the knee angle increased slightly (*p* < 0.001), the maximum hip moment decreased ($P_{hip}$ = 0.003), the motion range of the knee joints increased (*p* < 0.001), and the variation range experienced by the hip and knee power increased ($P_{hip}$ = 0.001, $P_{knee}$ < 0.001) in the state of “Exo. on” compared with the state of “Exo. off”. Due to the development of an unpowered exoskeleton in this article, its biggest and most unavoidable challenge is that the nodes for energy release and storage are very fixed. Therefore, it is necessary to evaluate the exoskeleton performance from the changes in joint power during the assistance phase. It can be clearly seen from Figure 11a that the first peak and the first valley of knee power were reduced by about 41.55% and 94.65%, respectively. In the design process of unpowered exoskeletons, the selection of springs was established without affecting normal gait. And our experimental results also indicate that, under normal gait, it does not affect the knee joint torque. It can be clearly seen from Figure 13a that the average joint moment of the knee was reduced by about 0.48% (LSD post hoc analysis: *p* = 0.920). This result indicates that wearing exoskeletons does not affect the knee moment. This precisely indicates that the designed exoskeleton did not affect normal walking gait and achieved the goal of assistance.

### 5.2. Plantar Pressure Tests

Healthy participants walked above the plantar pressure test board at normal walking speed, and tested the distributed plantar pressure test under the conditions of “No exo.”, “Exo. on” and “Exo. off”. The experimental data of dynamic plantar pressure distribution from the first contact of the heel with the ground to the first departure of the toe from the ground for the healthy participant were normalized to 101 data points. The changes in the plantar pressure center moving length, plantar force, plantar area, plantar average pressure, and plantar maximum pressure under three walking states were analyzed.

Figure 14 shows the variation in plantar pressure for the participant in walking tasks, which is a bar graph for the moving length of the plantar pressure center in three walking tasks. It can be clearly seen from the figure that for the state of “Exo. on”, the moving length of the plantar pressure center was less than that of “Exo. off”, and the moving length of the plantar pressure center was reduced by about 6.57% (*n* = 15 groups; one-way repeated-measures ANOVA; F = 4.982, *p* = 0.027; LSD post hoc analysis: *p* = 0.011). For the “Exo. off” state, when the moving length of the plantar pressure center was greater than that of “No exo.”, the moving length of the plantar pressure center increased by about 5.38% (LSD post hoc analysis: *p* = 0.038).

Figure 15a–d shows the variation diagram of the plantar force, area and pressure for the participant in the walking task. All data were time-normalized to 101 data points. Five groups were tested in each of the three states. The experimental data show that for the state of “Exo. on”, the plantar force, contact area, average pressure peak and maximum pressure peak were less than “Exo. off”. For the “Exo. off” state, the plantar contact area and the maximum pressure peak were greater than that of “No exo.”, while the plantar force and average pressure peak were less than that of “No exo.”. Through statistical analysis (one-way ANOVA), the experimental results were not significant (*p* > 0.05), which shows that the previous hypothesis is not tenable. Therefore, the wearing of the exoskeleton had no effect on the changes in the plantar force, area and pressure of participants. In the follow-up, we continue to study the effect of wearing the exoskeleton on plantar parameters.

In the dynamic distributed plantar pressure experiment, the COP length in the “Exo.on” state was reduced and compared with the “Exo.off” state (*p* = 0.027). However, compared with the state of “No exo.”, the state of “Exo. off” showed that the plantar contact area and the maximum pressure peak increased. On the one hand, by reducing the weight of the exoskeleton, we could reduce the load on lower human limbs when wearing the exoskeleton and continuously improve the assistance performance of the exoskeleton on the lower limb. On the other hand, further increasing static test experiments could verify whether the assisted exoskeleton can reduce the energy consumption of lower limbs from the physiological level through lower limb electromyography tests and metabolism tests.

### 5.3. Muscle Activity

A normalized EMG linear envelope of six tested muscles as a percentage of the gait cycle was used (heel to heel contact). As shown in Figure 16, these curves represent two different conditions: the “No exo.” (Black dotted line) and “Exo. on” (red dotted line) states. For the “No exo.” state, RF, VL, and SOL were more than the “Exo. on” state. Compared with the “Exo. on” state, SEM, TA, and PL were more than that of “No exo.”. The EMG test results show that when the participant wore the exoskeleton to assist walking, most EMG signals decreased slightly except for the semitendinosus, tibialis anterior, and peroneus longus muscle. This is because the exoskeleton is the external structure of the body, which causes additional muscle metabolism to perform work and consumes energy when walking.

We met with mixed success in our objectives. These results suggest that the tibialis anterior muscle can be assisted during the initial gait swing, although the semitendinosus muscle undergoes intense activity during the swing. Part of this reason may be caused by wearing problems during the experiment. In addition, there may be some differences in results due to differences in methods.

In conclusion, through the analysis of the experimental results, this can be used to evaluate the impact of an exoskeleton on human gait, plantar pressure, and muscle activation. Comparisons can be performed between the evaluation indexes of the proposed exoskeleton with the existing unpowered exoskeleton of the lower limbs, as shown in Table 4. For example, Banala, S.K. et al. [33] proposed a multi joint (hip and knee) gravity balanced leg orthosis. The angles of the hip and knee joints of healthy wearers increased, and the EMG activities of the rectus femoris and hamstring muscles decreased. Similarly, the motion range of the hip and knee angle in this paper increased, RF activity decreased, but SEM activity increased. Generally, in order to improve the assisted efficiency of exoskeleton, power is added to the unpowered exoskeleton for the operable electronic control system, clutch or variable shock absorber [34]. This is our next research direction regarding wearable-assisted exoskeletons.

## 6. Conclusions

In this paper, an unpowered knee exoskeleton is proposed, which can be used to reduce the energy consumption of lower limbs based on the energy compensation mechanism and achieve the purpose of assisted walking. The test prototype and engineering prototype were made using resin and aluminum alloy. The gait experiments and plantar pressure tests were carried out with an aluminum prototype to verify its effectiveness. In the experiment, the kinematic and dynamic data of the hip and knee, as well as plantar pressure characteristics, were collected and analyzed. The experimental results show that: (1) For the task of “Exo. on”, the variation range of the hip and knee power, knee angle, and hip moment was greater than that of the “Exo. off”, which increased by about 21.13% (*p* = 0.001), 39.07% (*p* < 0.001), 3.72% (*p* < 0.001), and 9.82% (*p* = 0.003), respectively. But during the assistance phase, the first peak and the first valley of knee power were reduced by about 41.55% and 94.65%m respectively. And an astonishing discovery was also made; the moving length of the plantar pressure center was less than that of “Exo. off”, which reduced by about 6.57% (*p* = 0.027). (2) For the “Exo. off” state, the range of the hip and knee power, knee angles, and hip moment were less than the “No exo.” state, which reduced by 21.60% (*p* = 0.001) and 21.69% (*p* < 0.001), 2.05% (*p* < 0.001), 13.61% (*p* = 0.003), respectively. But the moving length of the plantar pressure center was more than that of “Exo. off”, which increased by about 5.38% (*p* = 0.027). (3) The tibialis anterior muscle can be assisted during the initial gait swing, although the semitendinosus muscle experiences intense activity during the swing. It should be emphasized that the energy consumption and muscle activation of lower limbs needs to be further verified through subsequent experiments. It is believed that this assistance device could become a potential orthosis for stroke patients and used as an auxiliary rehabilitation device for patients with muscle weakness.

## Figures and Tables

**Figure 1 micromachines-14-01812-f001:**
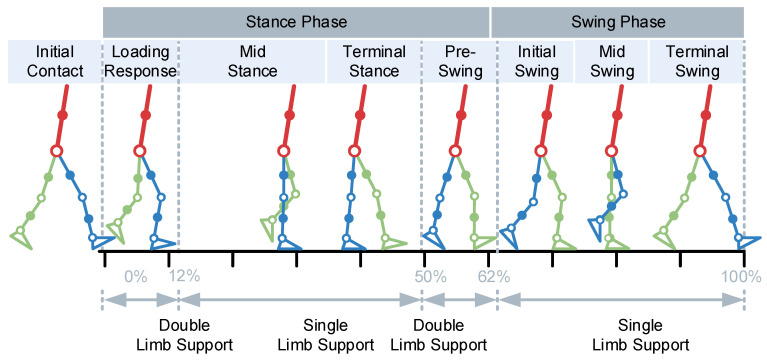
Typical gait phases.

**Figure 2 micromachines-14-01812-f002:**
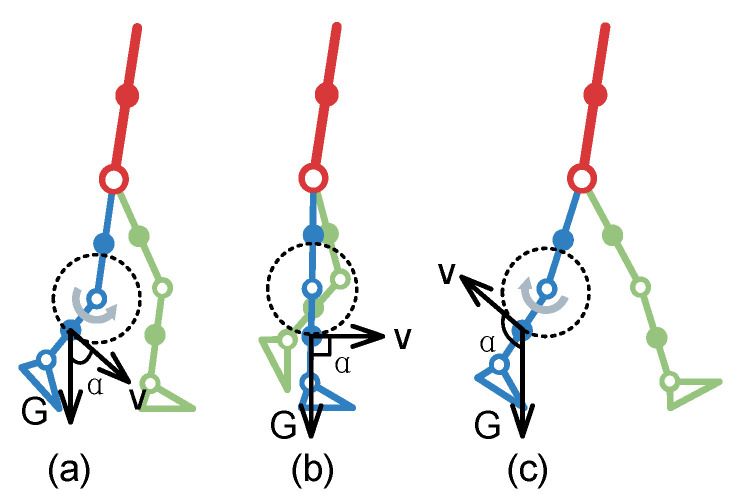
Schematic diagram of gravity balance state. The lower limbs perform (**a**) positive work, (**b**) no work and (**c**) negative work respectively.

**Figure 3 micromachines-14-01812-f003:**
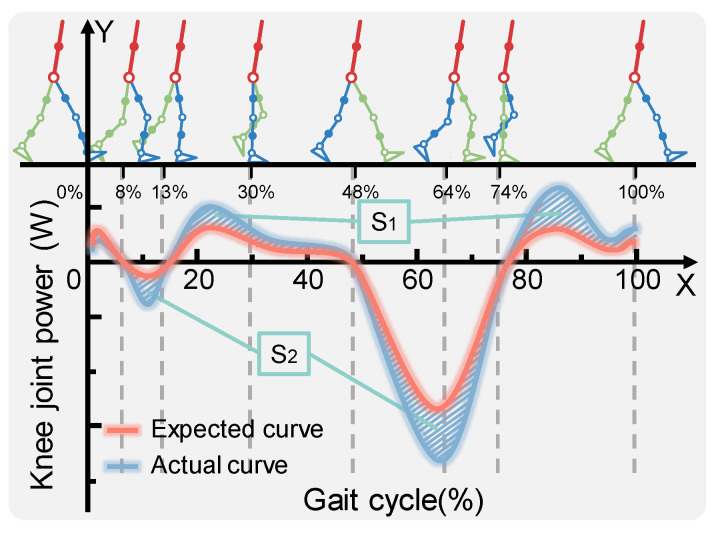
Fitting curve of the knee average power.

**Figure 4 micromachines-14-01812-f004:**
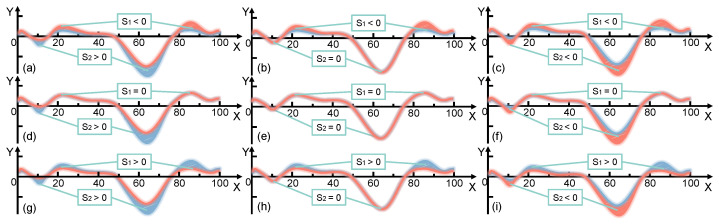
Power changes during assisted walking. Two conditions of joint power are described: an expected curve (red) and actual curve (blue). The area formed between the two curves is indicated using red (above the *X*-axis) and blue (below the *X*-axis) slashes respectively. When parameter $S_{1}$ is a negative number, $S_{2}$ is (**a**) positive, (**b**) 0 and (**c**) negative. When parameter $S_{1}$ is equal to 0, $S_{2}$ is (**d**) positive, (**e**) 0 and (**f**) negative. When parameter $S_{1}$ is a positive number, $S_{2}$ is (**g**) positive, (**h**) 0 and (**i**) negative.

**Figure 5 micromachines-14-01812-f005:**
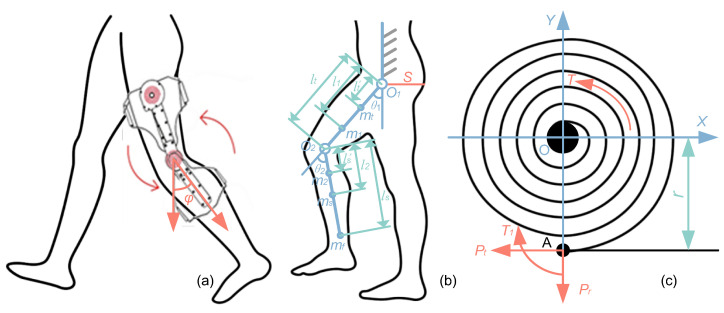
The human exoskeleton system. (**a**) State diagram of wearing a lower-limb exoskeleton under normal walking conditions (**b**) Geometric relationship of lower limb parameters (**c**) Force analysis diagram of scroll spring.

**Figure 6 micromachines-14-01812-f006:**
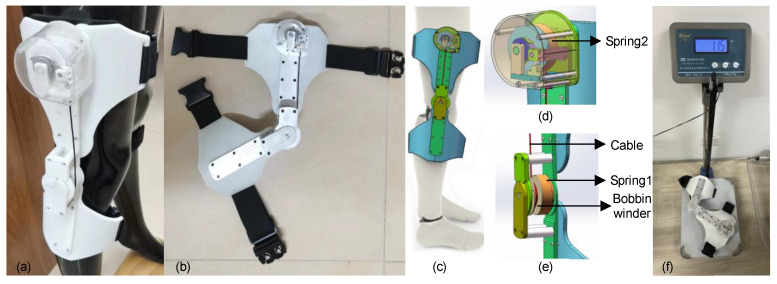
Structure of knee-joint exoskeleton. (**a**) The resin prototype. (**b**) The aluminum alloy prototype. (**c**) Man–machine coupling of 3D digital prototype. (**d**) and (**e**) provide the details of the device. (**f**) The device is on the electronic scale.

**Figure 7 micromachines-14-01812-f007:**
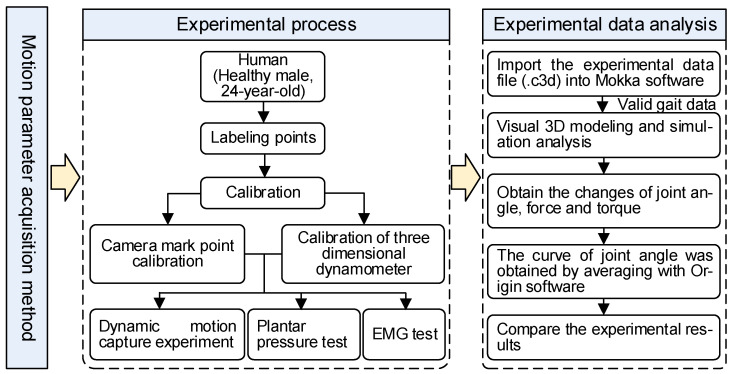
The flow chart of the experimental process and data analysis.

**Figure 8 micromachines-14-01812-f008:**
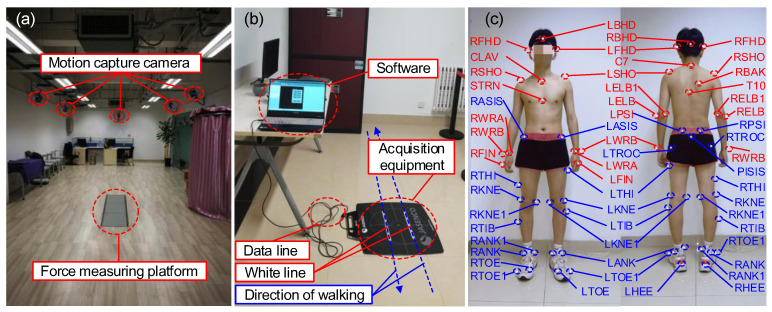
Experimental equipment and marking points. (**a**) Motion capture experiment. (**b**) Distributed plantar pressure test. (**c**) Marker points for full body.

**Figure 9 micromachines-14-01812-f009:**
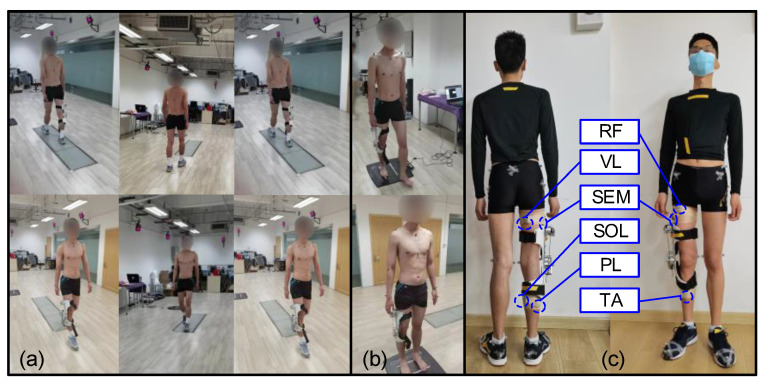
Experimental scene. (**a**) Dynamic capture and (**b**) Distributed plantar pressure test in three states. (**c**) Six tested muscle positions.

**Figure 10 micromachines-14-01812-f010:**
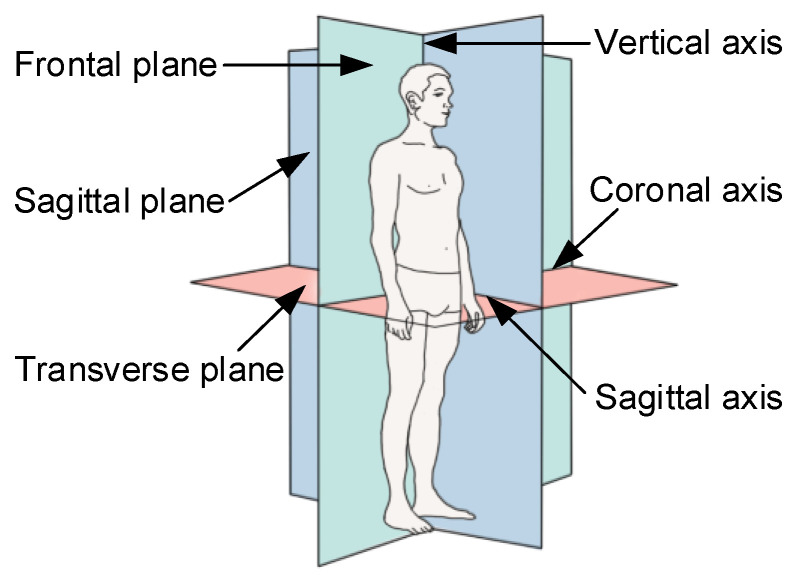
Human body reference plane and axis system.

**Figure 11 micromachines-14-01812-f011:**
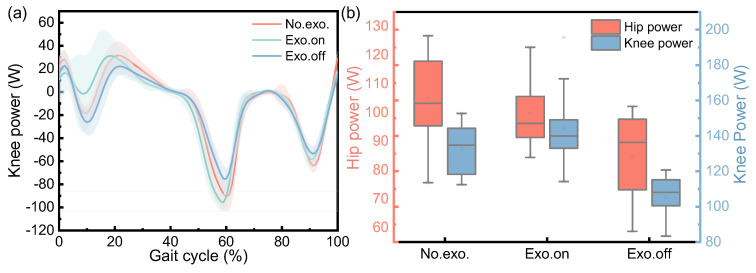
Variation in hip and knee joint power of the participant during the walking task. (**a**) The change in knee power for three conditions during the gait cycle. (**b**) The variation range in hip and knee power for three conditions.

**Figure 12 micromachines-14-01812-f012:**
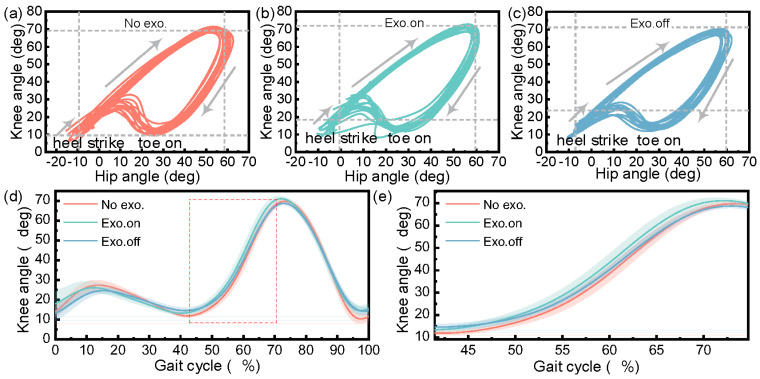
Participant’s hip–knee angle in the walking task. The hip–knee angle during the walking task for (**a**) “not wearing exoskeleton” (No exo.), (**b**) “wearing exoskeleton with assistance“ (Exo. on), and (**c**) “wearing exoskeleton without assistance” (Exo. off). (**d**) The change curve for the knee joint angle in a complete gait cycle. (**e**) Change in knee angle for 40~74% gait cycle (knee flexion to the maximum angle).

**Figure 13 micromachines-14-01812-f013:**
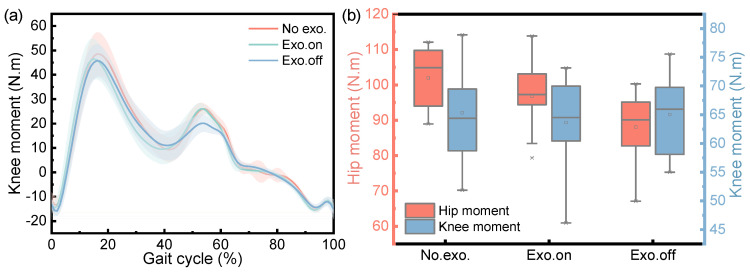
Variation in joint moment of participant during the walking task. (**a**) The change in knee moment for three conditions during the gait cycle. (**b**) The variation range of the hip and knee moment for three conditions.

**Figure 14 micromachines-14-01812-f014:**
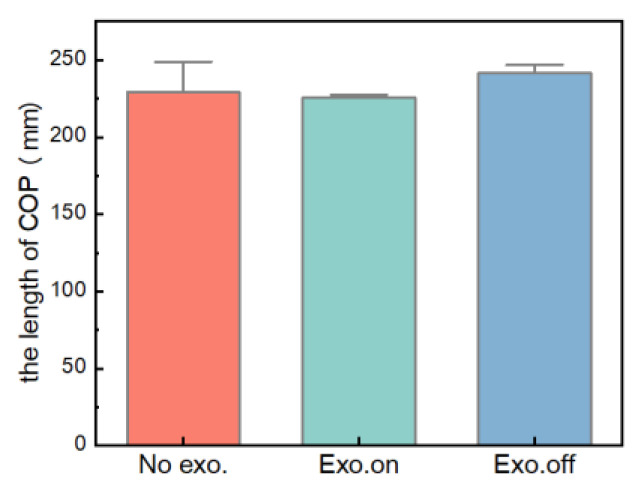
Comparison of the moving length of the plantar pressure center for the walking task.

**Figure 15 micromachines-14-01812-f015:**
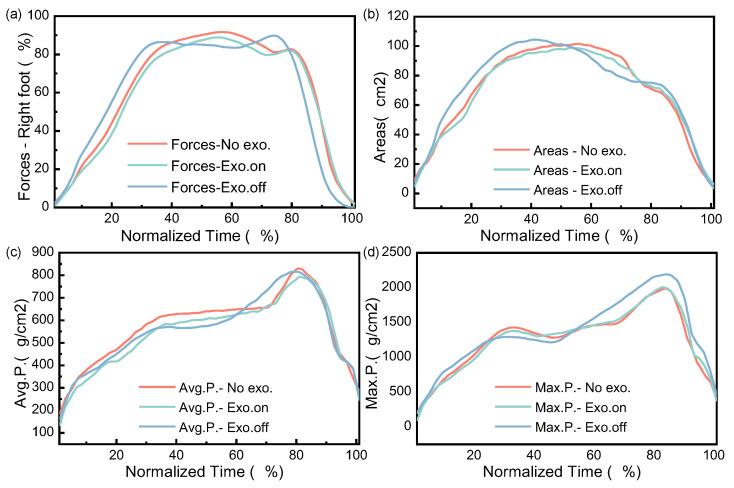
Variation in plantar force, areas and pressure in walking task. (**a**–**d**) Describe the changes in plantar forces, contact areas, average pressure, and maximum pressure, respectively. The time is normalized.

**Figure 16 micromachines-14-01812-f016:**
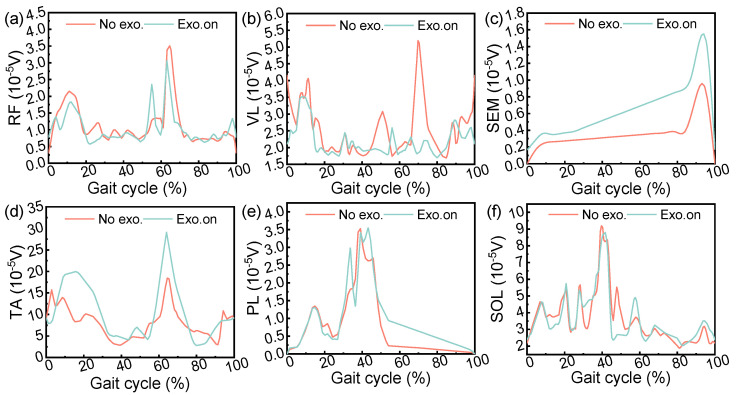
Muscle activation. These curves represent two different conditions: “No exo.” (red) and “Exo. on” (cyan). During the gait cycle (heel to heel contact), the active state of six tested muscles was analyzed, including (**a**) the rectus femoris, (**b**) vastus lateralis, (**c**) semitendinosus, (**d**) tibialis anterior, (**e**) peroneus longus, and (**f**) soleus muscle.

**Table 1 micromachines-14-01812-t001:** Six typical moments of knee joint.

Moment	0%	8%	30%	40%	64%	74%
Implication	Right plantar touch	Left toe-off	Right foot flat moment	Right leg straight	Right toe-off	Maximum flexion angle of knee joint

**Table 2 micromachines-14-01812-t002:** Assessment of the ability to capture energy.

Parameter 1	Parameter 2	Parameter 3	Hypothesis	Evaluation	Illustration
$P_{s1}<P_{s},P_{s2}<P_{r}$	$P_{n}=0,P_{m}>0$	$s_{1}<0,s_{2}>0$	$s_{2}>s_{1}$	Good	Figure 4a
			$s_{2}=s_{1}$	Medium	
			$s_{2}<s_{1}$	Poor	
$P_{s1}<P_{s},\;P_{s2}=P_{r}$	$P_{n}=0,P_{m}=0$	$s_{1}<0,s_{2}=0$	N/A	Poor	Figure 4b
$P_{s1}<P_{s},P_{s2}>P_{r}$	$P_{n}=0,P_{m}=0$	$s_{1}<0,s_{2}<0$	N/A	Poor	Figure 4c
$P_{s1}=P_{s},P_{s2}<P_{r}$	$P_{n}=0,P_{m}>0$	$s_{1}=0,s_{2}>0$	N/A	Good	Figure 4d
$P_{s1}=P_{s},P_{s2}=P_{r}$	$P_{n}=0,P_{m}=0$	$s_{1}=0,s_{2}=0$	N/A	Medium	Figure 4e
$P_{s1}=P_{s},P_{s2}>P_{r}$	$P_{n}=0,P_{m}=0$	$s_{1}=0,s_{2}<0$	N/A	Poor	Figure 4f
$P_{s1}>P_{s},P_{s2}<P_{r}$	$P_{n}>0,P_{m}>0$	$s_{1}<0,s_{2}>0$	N/A	Good	Figure 4g
$P_{s1}>P_{s},P_{s2}=P_{r}$	$P_{n}>0,P_{m}=0$	$s_{1}<0,s_{2}=0$	N/A	Good	Figure 4h
$P_{s1}>P_{s},P_{s2}>P_{r}$	$P_{n}>0,P_{m}=0$	$s_{1}<0,s_{2}<0$	$s_{1}>s_{2}$	Good	Figure 4i
			$s_{1}=s_{2}$	Medium	
			$s_{1}<s_{2}$	Poor	

**Table 3 micromachines-14-01812-t003:** Design parameters of the scroll spring.

Parameter	Value
Thickness h (mm)	0.8
Width b (mm)	10
$\mathrm{Torque}\;T_{2}$ (N·m)	1.689
$\mathrm{Diameter}\;\mathrm{of}\;\mathrm{mandrel}\;\mathrm{wound}\;d_{1}$ (m)	0.012
$\mathrm{External}\;\mathrm{diameter}\;d_{2}$ (m)	0.034
$\mathrm{Spring}\;\mathrm{coil}\;\mathrm{loose}\;\mathrm{ground}\;\mathrm{inner}\;\mathrm{diameter}\;D_{1}$ (m)	0.034
$\mathrm{The}\;\mathrm{inner}\;\mathrm{diameter}\;\mathrm{of}\;\mathrm{spring}\;\mathrm{box}\;D_{2\;}$(m)	0.047
$\mathrm{Number}\;\mathrm{of}\;\mathrm{turns}\;\mathrm{without}\;\mathrm{torque}\;n_{1}$ (cycles)	7.875
$\mathrm{Number}\;\mathrm{of}\;\mathrm{turns}\;\mathrm{after}\;\mathrm{spring}\;\mathrm{winding}\;n_{2}$ (cycles)	13.813
$\mathrm{Theoretical}\;\mathrm{working}\;\mathrm{revolution}\;n_{t}$ (turns)	3.060
$\mathrm{Effective}\;\mathrm{working}\;\mathrm{revolution}\;n_{e}$ (turns)	5.225
Energy storage level U (J)	16.230
Energy storage densityq (J/kg)	258.441
Maximum deformation angle φ (rad)	$(\pi \times 60{}^{\circ})/180^{{}^{\circ}}$
Spring stiffness K$\;(N\cdot m{rad}^{-1}$)	3
$\mathrm{Bearing}\;\mathrm{torque}\;T_{p}$ (N·m)	3.142
$\mathrm{Effective}\;\mathrm{energy}\;\mathrm{storage}\;\mathrm{level}\;U_{p}$ (J)	1.645
$\mathrm{Effective}\;\mathrm{energy}\;\mathrm{storage}\;\mathrm{density}\;q_{p}$ (J/kg)	29.193

**Table 4 micromachines-14-01812-t004:** A comparison of the evaluation indexes for different exoskeletons.

Study	Joint	$P_{h}$	$P_{k}$	$A_{h}$	$A_{k}$	$M_{h}$	$M_{k}$	RF	SEM	PF/GRF	L.COP	Met.
Ref. [33]	H.K.	--	--	↑	↑	--	--	↓	↓	--	--	--
Ref. [9]	H.K.A.	--	--	--	--	--	↓	--	--	--	--	↓
Ref. [7]	H.K.A.	--	--	--	--	--	--	--	--	↓	--	--
Ref. [19]	H.K.A.	--	--	--	--	--	--	--	↓	--	--	↓
Ref. [11]	H.K.	--	--	--	--	↓	↓	--	--	--	--	--
Refs. [23,24]	H.	--	--	--	--	--	--	--	--	--	--	↓
Ref. [35]	H.	--	--	--	--	↓	--	--	--	--	--	--
Refs. [21,28]	K.	--	--	--	--	--	--	--	--	--	--	↓
Ref. [36]	K.	--	--	--	--	--	--	--	--	↓	--	--
Ref. [20]	K.	--	--	--	--	--	--	↓	--	--	--	--
This work	K.	↑	↑	↑	↑	↑	--	↓	↑	--	↓	--

↑ represents an increase after wearing; ↓ represents a decrease after wearing; $P_{h}$: Hip power; $P_{k}$: Knee power; $A_{h}$: Hip angle; $A_{k}$: Knee angle; $M_{h}$: Hip moment; $M_{k}$: Knee moment; PF: Plantar force; GRF: Ground reaction forces; L.COP: Length of COP (center of pressure); Met.: Metabolic; H.K.: Hip and knee; H.K.A.: Hip, knee and ankle; H.: Hip; K.: Knee.

## Data Availability

Not applicable.

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
