# Peer review of "An Unpowered Knee Exoskeleton for Walking Assistance and Energy Capture"

_micromachines, 2023, doi:10.3390/mi14101812_

Round 1
Reviewer 1 Report
This work is interesting. Therefore, I recommended to be published, it after minor revision.
1) The abstract should be more informative.
2) In Figure 13 b unit on the Y axis is missing.
3) In Figure 15 figure arrangement is wrong specially a-b is missing.
4) In Figure 11 b unit on the Y axis is missing.
Reviewer 2 Report
This paper presents an unpowered knee exoskeleton for walking assistance and energy capture. The work is interesting. Following are some comments to improve this paper.
1: The advantages of the proposed exoskeleton compared with conventional exoskeletons are suggested to be included to highlight the contributions of this work.
2: Experimental results show that the the variation range of hip and knee power, the hip moment was greater under the task of “ Exo. on” compared with “Exo. off”. How these results verify the effectiveness of the proposed exoskeleton? Since the exoskeleton should be designed to reduce the human joints’ power and moment.
3: There are some self-contradiction contents in this manuscript. For example, in page 4, the words said “the red line represents the original average power curve Pa, and the blue line represents the expected average power curve Pe after assistance.” However, the red represents the expected curve and the blue represents the actual curve in the referenced figure 3.
4: The authors said in pages 3 and 4 that “ the energy capture device stores energy when the joint does positive work and is in the state of doing negative work. The energy capture device releases energy when the joint is doing negative work and is in the state of doing positive work.” This sentence is confusing. The unpowered exoskeleton usually stores energy when the joint does negative work.
The language should be carefully revised before resubmission. There are some self-contradiction contents between the words and the figures.
Reviewer 3 Report
The paper is very well written, and contributes an unpowered exoskeleton assisted device for the knee joint,which is used to reduce the energy consumption of lower limbs based on the energy compensation mechanism, so as to achieve the purpose of assisted walking. The effectiveness of the assisted device was verified by gait experiments and distributed plantar pressure tests with three modes. And the assist device can become a potential orthosis for stroke patients and can be used as an auxiliary rehabilitation device for patients with muscle weakness.
There are some problems, which must be solved before it is considered for publication. Relevant research background needs to be supplemented in INTRODUCTION. And you should cite the latest papers.
Round 2
Reviewer 2 Report
The authors have addressed my comments well. I think it can be accepted.